# Assessment of Elbow Proprioception with Inertial Measurement Units—Validity and Reliability Study

**DOI:** 10.3390/s25226826

**Published:** 2025-11-07

**Authors:** Szymon Stupnicki, Grzegorz Mulski, Łukasz Żytka, Jakub Kaszyński, Cezary Baka, Bartłomiej Lubiatowski, Przemysław Lubiatowski

**Affiliations:** 1Rehasport Clinic, 60-201 Poznan, Poland; szymon.stupnicki@rehasport.pl (S.S.);; 2Sport Trauma and Biomechanics Unit, Department of Orthopaedics, Traumatology and Hand Surgery Department, Poznan University of Medical Sciences, 61-545 Poznan, Poland; 3RSQ Technologies, 61-737 Poznan, Poland

**Keywords:** inertial measurement units, proprioception, joint position sense, elbow, motion analysis

## Abstract

Background: Inertial measurement units (IMUs) represent a relatively new and promising method for motion analysis. Their main advantages include small size and portability, combined with the use of advanced technologies. To date, few studies have investigated the application of these devices for proprioception assessment, and none have focused specifically on the elbow joint. Therefore, the aim of our study was to assess reliability and validate the protocol of elbow proprioception evaluation using inertial motion sensors. Methods: Twenty healthy participants underwent active and passive proprioception assessments based on joint position sense (JPS). Two researchers independently performed evaluation. The analyzed data was the error of reproduction of joint position (ERJP). IMU (RSQ Motion sensors) were used for angular joint position assessment and validated against Biodex System 4. Results: Inter-rater reliability for passive proprioception was good, with a Kendall’s coefficient of 0.77 (*p* < 0.05) for both RSQ Motion sensors and BIODEX, while active proprioception measured with RSQ Motion sensors showed slightly lower reliability (Kendall’s coefficient of 0.66, *p* < 0.05). Intra-rater reliability had similar results, with Kendall’s coefficients of 0.74 for passive BIODEX proprioception examination, 0.75 for passive RSQ Motion sensor testing and 0.65 for active proprioception (*p* < 0.05) measured with RSQ Motion sensors. The Bland–Altman plot revealed an equal distribution of results, which were within the limits of agreement (LoA). Conclusions: These results suggest proprioception assessment by JPS using inertial motion sensors is reliable and valid. It is an easy to use, light, portable, and inexpensive alternative for proprioception assessment, although further research in diverse clinical settings is needed.

## 1. Introduction

Proprioception is a crucial sensory function that ensures precise and coordinated human body movements. It allows individuals to perceive the position of their body parts in space and to integrate this information for accurate joint positioning and motor control [1,2]. This function is especially critical for athletes, where a high level of proprioception is needed for supreme performance [3,4].

Our research focuses on proprioception and novel methods of its assessment. It is well established that proprioceptive ability can be enhanced through targeted training or surgical restoration of function [3,4,5,6,7,8,9,10,11,12].

Conversely, it may be impaired by joint instability or surgical procedures with a significant degree of trauma [8,13,14,15,16]. These findings highlight the importance of prioritizing proprioceptive recovery during rehabilitation and minimizing unnecessary tissue damage whenever possible.

In addition to restoring range of motion, strength, and endurance, clinicians should place equal emphasis on the recovery of proprioception to ensure optimal performance and safety—particularly when patients are expected to perform complex movement patterns. The risk of re-injury is significantly higher following an initial trauma or surgery, especially in the early stages of recovery. At this stage, monitoring proprioception is essential, helping clinicians determine whether an individual is ready to return to sports or daily activities. A high level of proprioceptive function suggests that a patient is better prepared to resume physical activity safely, thus reducing the risk of further injury [17,18].

Beyond orthopedics and sports medicine, proprioception also has important implications in the field of neurology. Deficits in proprioception are frequently observed in patients with central nervous system disorders, such as cerebrovascular stroke and Parkinson’s disease [19,20,21,22,23]. In these populations, proprioceptive training and assessment have been shown to improve motor function, emphasizing the relevance of proprioception in neurorehabilitation [24,25].

Assessing joint proprioception remains a challenging task, as is the case with most sensory functions. It involves a subjective component based on the patient’s own perception [1,2]. One of the key concerns is the selection of an appropriate device to evaluate joint position accurately [26,27]. Even small discrepancies are clinically relevant; therefore, the measuring device should be minimal, lightweight, portable, and—most importantly—precise.

We have explored a novel approach to body motion analysis through the application of inertial measurement units (IMUs). These devices integrate subcomponents of the accelerometer, the gyroscope, and less commonly the magnetometer. IMUs utilize advanced mathematical algorithms, such as the Kalman filter or Madgwick filter to provide the real-time angular positioning of joints. These filters are used to integrate data from IMU subcomponents. In our study we used the Madgwick filter. This filter is specifically designed to estimate orientation using data from the accelerometer and gyroscope, without relying on magnetic measurements. By contrast, Kalman filter-based approaches typically require magnetometer data to correct yaw drift and provide an absolute directional reference [28,29]. Compared to widely used systems like the BIODEX, IMUs (e.g., RSQ Motion sensors) offer several advantages, including enhanced portability, a smaller size, and reduced cost—making them more accessible for use in clinical practice.

Methodology in proprioception research must be reproducible. One of the most common approaches involves the assessment of joint position sense and the accuracy of its reproduction by the patient. Although several methods have been described in the literature, few have undergone thorough validation and verification in the elbow joint [26,27,30]. Our research group has extensive experience in proprioception assessment, and, in previous studies, we have confirmed the accuracy and utility of single IMU devices for measuring shoulder and hip motion [31,32,33].

Therefore, the purpose of this study was to assess the reliability of a protocol for evaluating elbow proprioception using inertial motion sensors and to validate the use of these sensors in comparison with a well-established reference device—the Biodex dynamometer [34].

## 2. Materials and Methods

### 2.1. Participants

Twenty healthy adult volunteers were recruited for the study (aged 20–25 years; 4 females and 16 males). None of the participants had a history of upper extremity trauma or surgery. Additionally, they reported no symptoms related to the elbow or any other musculoskeletal complaints within the 12 months preceding the evaluation, and no neurological conditions were present. To eliminate any potential learning effect on the results, all participants were unfamiliar with the proprioceptive assessment procedures prior to the study.

### 2.2. Instrumentation

The complete setup for JPS (joint position sense) testing included two systems:RSQ Motion IMU sensors (RSQ Motion, RSQ Technologies, Poznań, Poland), andBIODEX System 4 (Biodex Medical Systems, 49 Natcon Drive, Shirley, NY, USA), considered the gold standard for comparison.

IMU (Inertial Measurement Unit)—RSQ Motion:

This system comprises the following components: a motion sensor, an elastic band for sensor stabilization, a handheld clicker used by the participant to indicate perceived joint position, and a mobile device (tablet) connected via Bluetooth to both the sensor and the clicker for real-time data collection (Figure 1). The device has an angular measurement accuracy of 0.15° [31]. Key RSQ Motion sensor specifications are described in table below (Table 1).

BIODEX System 4:

The Biodex System 4 is a comprehensive system consisting of a stable frame, main drive unit, adjustable chair, and computer interface [34]. It is primarily designed and well evaluated for isokinetic strength testing but also offers reliable angular position measurement and a controlled motion environment. The previous generation of this device has been validated for all these purposes. [35]. Besides the isokinetic testing protocol, it has a passive proprioception testing protocol (passive JPS), which was the subject of our work. The manufacturer reported an angular measurement accuracy of 1° [34].

### 2.3. Participants’ Positioning

Each participant was comfortably seated in the Biodex System 4 chair. The tested upper limb was positioned on the dedicated Biodex lever, with the elbow aligned at shoulder height and the lateral epicondyle placed next to the base of the lever; the forearm was in a neutral position.

The inertial motion sensor was securely attached between the styloid processes of the ulna and radius using an elastic band to ensure stable positioning. The clicker was held in the participant’s free contralateral (not examined limb) hand. Before each passive and active JPS assessment, the device was recalibrated. It involved placing the sensor on the ground and zeroing it in this position.

To eliminate the influence of visual feedback, all participants were blindfolded during testing [36] (Figure 2). The assessment was conducted in a quiet room to minimize external distractions.

The initial active and passive JPS assessments began with a trial comprising two single-position recordings, allowing the participant to familiarize themselves with the use of the devices.

### 2.4. Active Joint Position Sense Evaluation Protocol

Before each test, participants received a brief explanation of the procedure. The Biodex lever was initially positioned at 90°, after which the researcher moved it to one of the target reference angles—50°, 70°, or 110° of elbow flexion—where it was held for five seconds (Figure 3). During this time, the participant was instructed to memorize the joint position and press the clicker to mark it. Subsequently, the participant actively returned the limb to the starting position (90°) and then attempted to actively reproduce the target angle. Once the participant believed the desired position was reached, they pressed the clicker again to confirm the achieved angle. The reference angles were tested in the following order: 70°, 50°, and 110°.

The 50° and 70° targets were reached by actively extending the elbow from the initial 90° position, while the 110° target was reached through active flexion. Each target angle was tested three times on both limbs.

The Biodex lever was used solely to control the plane of motion, thereby minimizing compensatory movement from the shoulder and trunk muscles. In this active JPS assessment module, only the RSQ Motion sensor was used for data acquisition. The Biodex system was not utilized in this part of the study, as it is only capable of evaluating passive joint position sense.

### 2.5. Passive Joint Position Sense Evaluation Protocol

As with the active protocol, participants first received a standardized explanation of the procedure. The Biodex lever was initially positioned at 90° and then moved manually by the researcher to the selected target angle (50°, 70°, or 110° of elbow flexion), where it was held for five seconds (Figure 3). The participant was instructed to memorize the joint position and pressed the clicker to confirm memorization. The lever was then returned to the starting position (90°). Next, the Biodex system passively moved the limb at a slow and controlled speed toward one of the target angles. The angular velocity of the Biodex arm was 5 degrees per second. The participant was asked to indicate when they perceived that the joint had reached the previously memorized position by saying “stop” and simultaneously pressing the clicker. The researcher then immediately halted the movement of the lever, and the current joint angle was recorded both by the Biodex system and RSQ Motion sensor.

This procedure was repeated three times for each target angle (50°, 70°, and 110°) on both limbs in the same order as in active JPS evaluation protocol (70°, 50°, and 110°). During the passive JPS evaluation, data were collected simultaneously using both the Biodex system and the RSQ Motion sensors, allowing for direct comparison between the two measurement methods.

### 2.6. Data Collection and Statistical Analysis

Each participant underwent three rounds of testing in both the active and passive joint position sense (JPS) modules. Two of these rounds were conducted on the same day by different examiners to assess inter-rater reliability (reproducibility). A third round was performed one week later by one of the original examiners to assess intra-rater reliability (repeatability).

Analyzed data was the mean error of reproduction of joint position (ERJP)—in other words the absolute difference between the reference angle and angle reproduced by the patient. In total, 2160 measurements were performed (20 participants × 2 modules × 3 angles × 3 repetitions × 2 limbs × 3 sessions). Four measurements were excluded due to improper sensor calibration (two from the active module and two from the passive module).

The normality of the data distribution was assessed using several statistical tests, including the Lilliefors test, the Shapiro–Wilk test, and the D’Agostino–Pearson test. All tests indicated that the data did not follow normal distribution. In addition, skewness was evaluated, and Q–Q (quantile–quantile) plots were inspected to visually confirm deviations from normality. Based on these results, non-parametric statistical methods (dedicated for non-normally distributed data), were applied in the subsequent analyses. Kendall’s coefficient of concordance (W) was used to evaluate both inter- and intra-rater reliability (see Figure 4). Its interpretation is analogous to Intraclass Correlation Coefficient (ICC), with possible values ranging from 0 (no agreement) to 1 (complete agreement) [37,38,39]. Moreover, the Wilcoxon test for paired samples was employed as a non-parametric method to assess if there are differences in inter- and intra-rater groups. For the same purpose we defined the mean absolute value of ERJP difference.

To assess the agreement between the RSQ Motion sensors and the BIODEX system in the passive JPS module, a Bland–Altman plot was conducted [40,41]. Additionally, the mean absolute value of ERJP difference between RSQ Motion sensors and the BIODEX was checked.

## 3. Results

### 3.1. Inter-Rater Reliability

Kendall’s coefficient of concordance indicated a good level of inter-rater reliability for passive proprioception measurements taken with the BIODEX system, yielding a value of 0.77 (*p* < 0.05). Surprisingly, the passive proprioception measurements obtained using RSQ Motion sensors demonstrated identical reliability, with a Kendall’s coefficient of 0.77 (*p* < 0.05). In the case of active proprioception measured by RSQ Motion sensors, Kendall’s coefficient was moderate, at 0.66 (*p* < 0.05). The Wilcoxon test for paired samples revealed no statistically significant differences between raters in both modules when using the IMUs, nor in the BIODEX system measurements. Plots additionally illustrate agreement between raters (Figure 5).

The mean absolute difference in ERJP measurements (pooled data of positions and sides) performed by different raters for passive proprioception testing by the BIODEX system were 1.5° (95%CI: 1.2–1.7; SD: 1.3), for the passive testing of the RSQ Motion sensors 1.7° (95%CI: 1.4–1.9; SD: 1.5), and for the active module of the RSQ Motion sensors it was 1.6° (95%CI: 1.3–1.8; SD: 1.3).

### 3.2. Intra-Rater Reliability

Intra-rater reliability was similarly good for passive RSQ Motion testing. It was almost the same (although defined as moderate) for BIODEX and moderate for active testing. For the BIODEX system, Kendall’s coefficient of concordance was 0.74 (*p* < 0.05), while for the RSQ Motion sensors, the value was 0.75 (*p* < 0.05) for passive proprioception and 0.65 (*p* < 0.05) for active proprioception. The Wilcoxon test for paired samples showed no significant differences between measurements taken by the same rater across BIODEX system and RSQ Motion active proprioception evaluation. In the passive proprioception module there were statistical differences, but this remains clinically insignificant with a mean difference less than 0.5° (and less than 0.5° 95%CI differences). Plots additionally illustrate agreement among the same rater (Figure 6).

The mean absolute difference in measurements of ERJP performed by the same rater for BIODEX testing was 1.4° (95%CI: 1.1–1.7; SD: 1.3), for the passive module of the RSQ Motion sensors 1.8° (95%CI: 1.5–2.1; SD: 1.5), and for the active module of the RSQ Motion sensors 1.8° (95%CI: 1.5–2.2; SD: 1.7).

### 3.3. Agreement Between RSQ Motion Sensors and BIODEX

The Bland–Altman plot demonstrated that the majority of the results (blue dots) were within the limits of agreement defined by LogA (two most peripheral red dashed lines). Results (blue dots) are represented as errors between sensor measurement and BIODEX measurement.ERROR=θ_IMU−θ_Biodex ;(θ_IMU and θ_Biodex representing theangles measured by the IMU and Biodex, respectively),

They were distributed on the X axis according to arithmetic mean.$$MEAN=\frac{\left(\theta \text{\_}IMU+\theta \text{\_}Biodex\right)}{2}$$

The limits of agreement (LogA) are calculated as the mean difference ± 1.96 times the standard deviation of the differences (Figure 7). Additionally, the results were evenly distributed above and below the mean difference line (blue middle line), indicating that the device neither consistently over- nor underestimated the angles when compared to the other [41]. The mean absolute value of ERJP difference between RSQ Motion sensors and the BIODEX system was 0.9° (95%CI: 0.8–1; SD: 0.7), indicating a high level of agreement between the two measurement methods.

## 4. Discussion

Proprioception is a fundamental component of sensorimotor control, containing the perception of joint position, movement, and force. Among these, joint position sense (JPS) represents only one aspect of proprioception, primarily focusing on the conscious awareness of static joint angles. It should be emphasized that JPS alone does not reflect the full spectrum of the proprioceptive function, which is very complex and also consists of the sensation of dynamic joint movement (kinesthesia) [1,2]. Assessment methods for proprioception can be categorized into passive and active proprioception tests. Passive assessments involve movement of the limb without voluntary muscle activation, reflecting perception capacities. In contrast, active assessments require the participant to perform voluntary movements, thus integrating proprioceptive feedback with motor control mechanisms. This distinction is clinically relevant, as active proprioception is more representative of the real-life motor function [1,2]

Several types of devices, which are able to evaluate JPS are available on the market. Commencing with least advanced manual goniometers, they are inexpensive but lack precision and are highly operator-dependent [42,43]. Propriometers, previously used in our center, offer good accuracy but are large, wired, and impractical for routine use [44]. Biodex systems, regarded as the gold standard, are at the same time expensive, bulky, and limited to specialized facilities [27,45]. Specialized robots have similar characteristics [46,47]. Finally, there are also optoelectronic motion capture systems (e.g., Vicon), which provide highly accurate kinematic data but are cost-prohibitive, complex, and require trained personnel [48]. Among these, inertial measurement units seem to be a promising alternative, as they meet certain technical criteria. They are small in size, easy to apply and portable. Therefore, there is a clear possibility of using them in any setting, such as a medical office or a playing field. Moreover, inertial motion sensors have a relatively low cost. Compared to professional optometric systems and BIODEX, whose costs range from tens of thousands of euros and may, in some cases, approach one hundred thousand euros, the basic sensor setup used in this study costs approximately one thousand euros. All these features combined together have potential for their broad expansion in the future.

Most previous studies on proprioception have focused on the shoulder and knee joints, likely due to their greater range of motion and relevance in postural and gait stability. In contrast, the elbow joint has received limited attention in proprioceptive research. Nonetheless, the elbow plays a critical role in many functional and sport-specific tasks and is frequently affected by overuse injuries, trauma, and degenerative changes. Moreover, the elbow is often subject to surgical interventions—including minimally invasive procedures such as arthroscopy—where postoperative proprioceptive deficits may affect clinical outcomes [6,13,14,49,50,51,52]. Therefore, reliable and accessible tools for evaluating elbow proprioception are warranted in both diagnostic and rehabilitative contexts.

So far, to our knowledge, there is a lack of studies about inertial motion sensors in the context of joint proprioception measurement, although there is evidence in the literature showing that these types of devices perform well in range of motion assessment [31,32,33,53,54]. They can also be successfully used for balance and motion tracking [55,56,57]. This opens an opportunity to further expand their use in proprioception evaluation. However, the requirements for the use of IMUs for this purpose may be higher than those for assessing the range of motion. The measurement values of ERJP are significantly smaller than the range of motion values. Moreover, professional athletes are the ones who possess a superior level of proprioception sense, and small disturbance in this ability may affect their performance [19,20]. Therefore, future sensors that could be used for assessing proprioception should be highly accurate. For precise movements, devices with an accuracy of less than 1°, or even 0.1° are recommended [8,58]. RSQ Motion sensors met these requirements with an accuracy below 0.15° during validation with a robot arm [31]. Most studies related to body position assessment are based on the usage of devices containing accelerometer and gyroscope technology only [28,29,55,59]. Despite these two components, the RSQ Motion sensor contains an additional magnetometer, making it more advanced. This fact could prove crucial for its precision. Thanks to its design, the RSQ Motion sensor has the potential to meet these potentially higher requirements for JPS testing.

The RSQ Motion sensor testing demonstrated good inter- and intra-rater reliability for passive proprioception evaluation, similar to the widely used BIODEX system. The direct comparison between the BIODEX and RSQ Motion sensor testing showed no statistically significant differences, with a mean absolute difference in less than 1° between these devices. This is less than the BIODEX measurement interval and accuracy. The results of the passive proprioception evaluation with RSQ Motion sensors showed equal reliability to examination by gold standard BIODEX, suggesting that these portable, inexpensive sensors could be a reliable alternative in clinical settings. Active proprioception measurement was slightly worse; however, this module was more demanding to perform by the participants in our study settings. Patients were influenced by the BIODEX lever weight, because in this module, in contrast to the passive module, participants had to move the lever by overcoming its mass. From our observations, patients found it difficult to keep the levers stationary at the target point while replicating the angles. Minor movements at the moment of data reading by the RSQ Motion sensors likely affected the lower reliability in this module. This could be attributed to protocol-related factors, which may account for the lower reliability observed in active JPS testing. Additionally, we cannot rule out a further decrease in reliability in the active condition due to the greater cognitive and motor demands to actively reproduce a joint angle. Considering both of these factors, it is unlikely that the lower reliability in active JPS assessment was caused by the IMU system itself. Summarizing these difficulties, the results of active testing were still satisfactory.

As a main limitation, it is important to note that this study was conducted in a laboratory environment, in controlled settings that allowed for a direct comparison between RSQ Motion sensors and the BIODEX system. In both modules the reference angles were tested in the following order: 70°, 50°, and 110°. In our opinion, this order did not influence the study outcomes; however, we did not directly examine whether randomizing the sequence of angles would produce different results compared to the fixed order used in this study. It is worth adding that in the passive module, sensing limb position while simultaneously preparing to press a button introduces potential cognitive load. Participants did not report any difficulties in simultaneously performing the proprioception tasks and pressing the button, and the researchers did not observe any such issues. Therefore, we assume that cognitive load did not significantly affect the study outcomes. It should be noted, however, that the participants were young, healthy individuals without conditions affecting cognitive function.

Further research is needed to validate the performance of these inertial motion sensors in more diverse settings, particularly outside of the BIODEX chair and without the use of a lever. Moreover, including only healthy participants may be seen as a limitation. However, we consider this an essential step, as the reliability and validity of the method should first be tested in a homogeneous group of healthy individuals before being applied in clinical populations. Although the sample size of 20 participants may seem limited, it was adequate for our study design. The group was homogeneous in terms of proprioception, which reduced variability and ensured representativeness. In addition, 2160 measurements were collected, and all tests reached statistical significance (*p* < 0.05). Therefore, we consider the sample size sufficient for the objectives and methodology of our JPS assessment. Furthermore, it is possible that we achieved favorable results because of the fact that the elbow is a constrained, hinged joint, with relatively lower degrees of freedom, where movement is mainly conducted in one plane (elbow flexion). For this reason, in our study, a single sensor was sufficient to adequately measure the JPS. Ongoing studies should focus on assessing other joints, with more complex movement patterns, where using more than one sensor would probably be a rationale for proper JPS evaluation. This would state whether RSQ Motion sensors can be considered a universal tool for proprioception measurement across different body parts. It is important to add that we cannot fully exclude the learning effect in study participants. That explains why results from examinations after an interval of time are slightly better, but not clinically relevant (intra-rater reliability plot: Figure 6). With regard to potential technical confounders, several factors deserve consideration. Sensor drift can lead to measurement errors. However, in the course of our study, only a few minor episodes of device recalibration were observed. The measurements were conducted under stable and controlled environmental conditions (similar temperature, humidity, and a closed indoor setting), which likely minimized the impact of sensor drift. Concerning soft-tissue artifacts, the sensors were placed over prominent bony landmarks (the styloid processes of the ulna and radius). This ensured stable positioning and minimized the thickness of underlying soft tissue, thereby reducing the likelihood of soft-tissue-related errors.

It is worth mentioning that fMRI may enhance proprioception evaluation as there are specific signs of brain activity in peripheral proprioception deficit [60,61]. Combining peripheral JPS assessment and central nervous system imaging may be rationale in the future. Ongoing developments in proprioceptive assessment are likely to focus on motion monitoring during daily activities and sports participation. Wearable IMU systems integrated with mobile health platforms may enable real-time feedback and longitudinal tracking of proprioceptive function in both healthy individuals and clinical populations. Such advancements have the potential to transform rehabilitation paradigms by providing precise, individualized metrics of sensorimotor performance and enabling remote, data-driven therapeutic interventions.

## 5. Conclusions

Our protocol of elbow proprioception assessment based on JPS using IMUs (RSQ Motion) demonstrated good validity and reliability with performance comparable to the Biodex system in passive joint position sense testing. Active testing showed moderate reliability but remained clinically acceptable. Given portability, low cost, and ease of use, IMUs represent a practical alternative for proprioception evaluation. Future research should confirm applicability of the assessment protocol in clinical and sports settings.

## Figures and Tables

**Figure 1 sensors-25-06826-f001:**
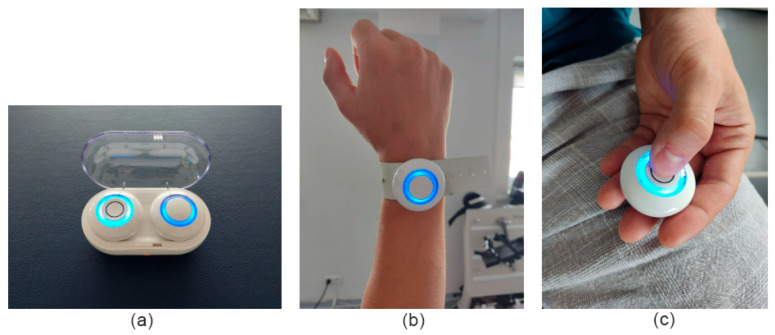
(**a**) Inertial motion sensor (RSQ Motion) and clicker charging box, (**b**) sensor fixed by elastic band, and (**c**) clicker pressed by participant to signalize the angular position.

**Figure 2 sensors-25-06826-f002:**
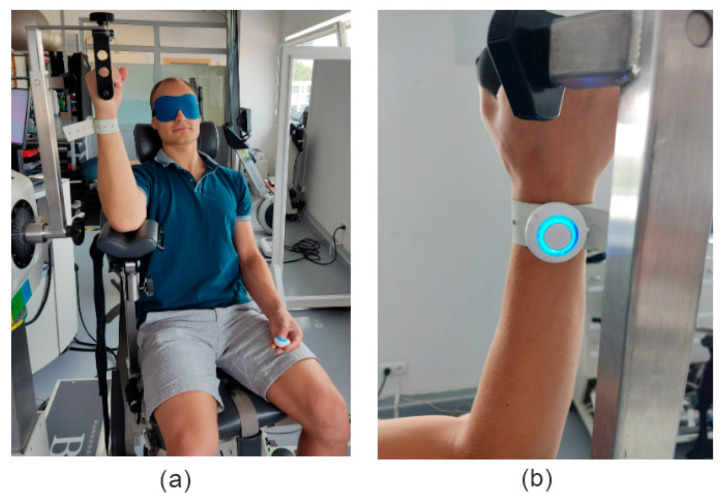
(**a**) Participant positioning on the BIODEX chair and (**b**) sensor placement next to styloid processes of ulna and radius.

**Figure 3 sensors-25-06826-f003:**
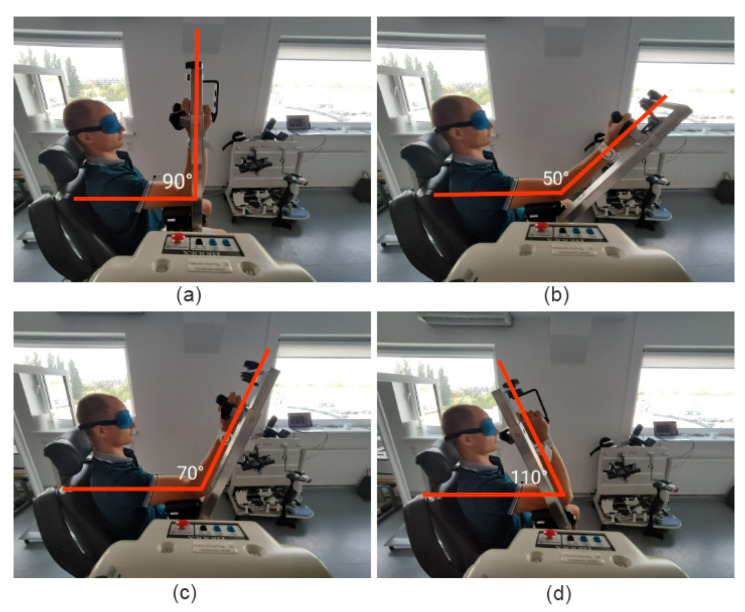
(**a**) Starting position: 90° of elbow flexion. Assessed angular positions: (**b**) 50° elbow flexion, (**c**) 70° elbow flexion and (**d**) 110° elbow flexion.

**Figure 4 sensors-25-06826-f004:**
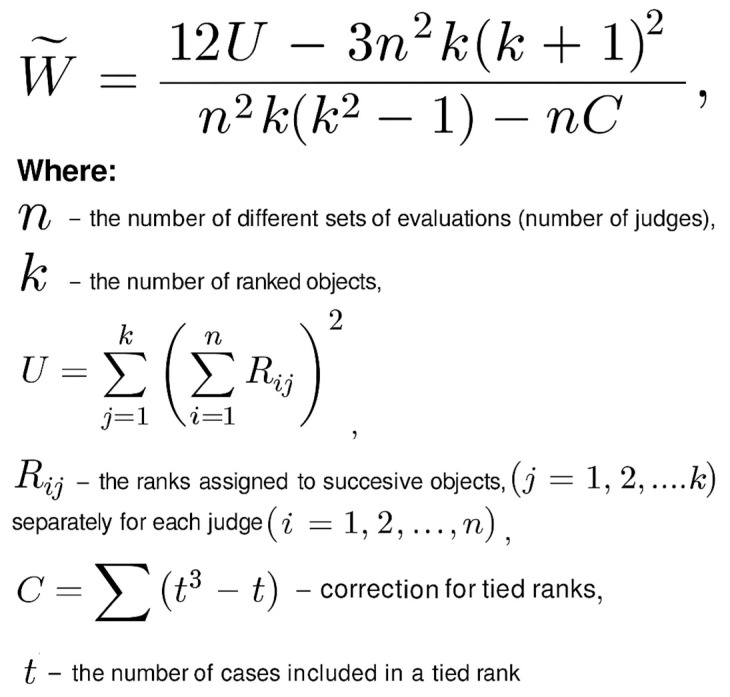
Formula for calculating inter and intra rater reliability with Kendall’s coefficient of concordance (W).

**Figure 5 sensors-25-06826-f005:**
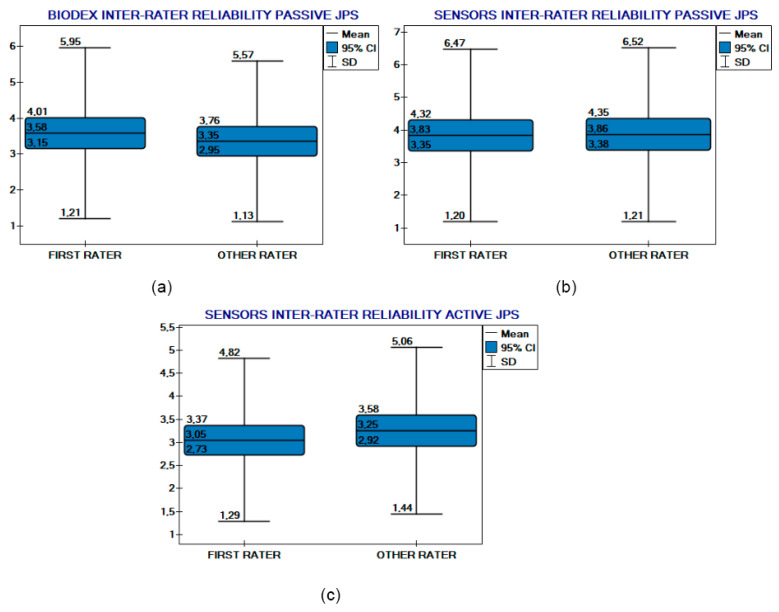
Inter-rater reliability comparison plots. (**a**) BIODEX inter-rater reliability (passive JPS). RSQ Motion Sensors (**b**) passive and (**c**) active JPS inter-rater reliability.

**Figure 6 sensors-25-06826-f006:**
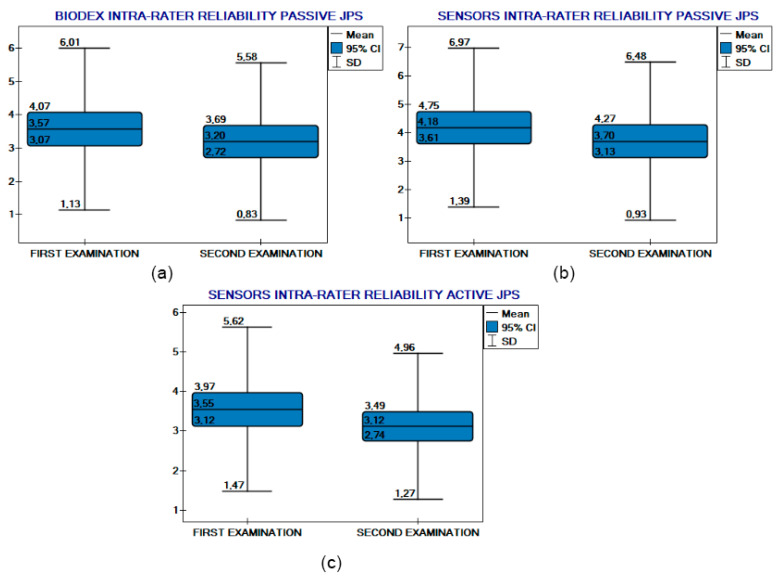
Intra-rater reliability comparison plots. (**a**) BIODEX intra-rater reliability (passive JPS). RSQ Motion Sensor (**b**) passive and (**c**) active JPS intra-rater reliability.

**Figure 7 sensors-25-06826-f007:**
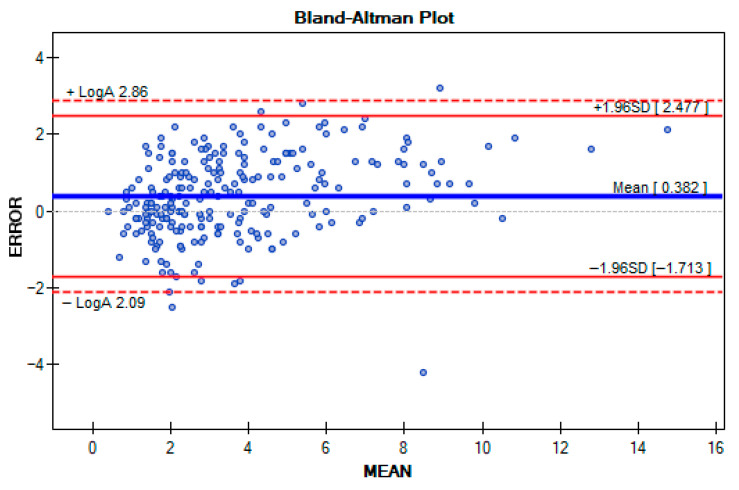
Differences between BIODEX and RSQ Motion sensors illustrated by the Bland–Altman Plot.

**Table 1 sensors-25-06826-t001:** Key RSQ Motion Sensor Specifications provided by the manufacturer.

Sensor Feature	Specifications
**Components**	accelerometer, gyroscope, magnetometer
**Data fusion method**	complementary filter (Madgwick)
**Magnetometer**	disabled to avoid interference from the Biodex
**Accelerometer trust value**	0.1
**Calibration process**	performed by the manufacturer (RSQ) using high-precision reference equipment (e.g., industrial robot); calibration coefficients determined: cross-alignment, scale factor, and offset
**Orientation computation**	orientation calculated from raw accelerometer and gyroscope data using sensor fusion (Madgwick filter)
**Sampling** **frequency/synchronization**	100 Hz
**Joint angle calculation**	Joint angles were obtained by comparing the orientation readings from two adjacent segments and calculating the relative difference between them
**Run time on single battery charge**	up to 8 h
**Battery life**	500 charging cycles
**Power source**	Rechargeable lithium-ion battery
**Charging voltage and current**	5 V, 90 mA

## Data Availability

The original contributions presented in this study are included in the article. Further inquiries can be directed to the corresponding author(s).

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
