# Peer review of "Assessment of Elbow Proprioception with Inertial Measurement Units—Validity and Reliability Study"

_sensors, 2025, doi:10.3390/s25226826_

Round 1
Reviewer 1 Report
Comments and Suggestions for Authors
Review on paper
Assessment of elbow proprioception with Inertial Measurement Units- Validity and Reliability study
Paper overview
The manuscript presents a study regarding the use of inertial measurement units (IMUs) for assessing elbow joint proprioception, focusing on both validity and reliability – which to my best knowledge is one of the first such studies on this joint. The work is well done and relevant, considering the increasing adoption of wearable sensors in clinical and sports medicine with the continuous developments in robotic assisted rehabilitation procedures. While IMUs have been previously validated for range of motion assessments in other joints, their application for elbow proprioception evaluation remains underexplored, which, as I mentioned before gives this research a higher degree of novelty. The study is well designed, uses appropriate statistical methods, and is clearly structured. However, some aspects of methodology and discussion would benefit from clarification and expansion. I will continue the review with some detailed explanations to suggest some minor improvements.
While I have acknowledged one of the first usage of IMUs for the assessment of proprioception one limitation that has to be mentioned is the use of exclusively healthy patients in the study. However, this is more a remark as working in this field for many years the validation has to be done firstly on healthy population and once everything is setup properly, the next step is to move towards patients with strong clinical support.
Maybe a discussion about the size of the population is worth mentioning as the sample size is not very large, and it should be discussed whether this number (20) might introduce some limitations.
I would recommend, in Discussions, to expand a bit the potential technical confounders like sensor drift, soft-tissue artifacts, magnetometer interference and add some more details on the robot arm setup.
The figures are of good quality and readable. If possible, as I magnified them on my screen, maybe the export in a higher resolution would increase even more the quality (but they are quite fine as they are).
As a personal curiosity and maybe worth mentioning in the paper, the blindfolding of the participants is motivated (to avoid distractions), but I was curious how was this perceived in terms of comfort and emotional stress. Of course, in a clinical setup, after the solution is validated the blindfolding is not necessary anymore.
As a final conclusion, I did enjoy reading the paper and I consider it a valid contribution in the field of robotic (technology) assisted rehabilitation. For this first draft I recommend the acceptance of paper with minor changes.
Author Response
Comments 1: While I have acknowledged one of the first uses of IMUs for the assessment of proprioception, one limitation that has to be mentioned is the use of exclusively healthy patients in the study. However, this is more a remark as working in this field for many years the validation has to be done firstly on a healthy population and once everything is set up properly, the next step is to move towards patients with strong clinical support.
Response 1: Thank you for pointing this out. We agree with this comment. Therefore, we have mentioned in discussion, under the limitation paragraph, that our study included only healthy patients. Changes can be seen on page 13, lines 378-381.
Comments 2: Maybe a discussion about the size of the population is worth mentioning as the sample size is not very large, and it should be discussed whether this number (20) might introduce some limitations.
Response 2: Thank you for pointing this out. We agree with your comment. Accordingly, we have mentioned in the discussion section, under the limitations paragraph, that our study included a relatively small number of patients (20). We have also explained why this sample size was sufficient in the context of our objectives and methodology. The changes can be found on page 13, lines 381-386.
Comments 3: I would recommend, in Discussions, to expand a bit the potential technical confounders like sensor drift, soft-tissue artifacts, magnetometer interference and add some more details on the robot arm setup.
Response 3: Thank you for pointing this out. We agree mostly with your comment. Therefore, we have mentioned in the discussion section, under the end of limitations paragraph, all these technical confounders. Moreover we described how they were minimized in the context of our study. The changes can be found on page 14, lines 396-404. Magnetometer interference was not the case in our study as it was disabled to avoid BIODEX interference (stated by sensors provider/manufacturer) - key sensor features can be seen in new Figure 1 on page 4. Details on the robot arm setup, mentioned in discussion, can be seen in the reference article. In this article figures were used primarily to describe this setup. As it is technically challenging to briefly describe the robot arm configuration without figure support, we would prefer to retain the current presentation and not implement this suggested change. Readers can successfully obtain information about the robot arm setup from the referenced article.
Comments 4: As a personal curiosity and maybe worth mentioning in the paper, the blindfolding of the participants is motivated (to avoid distractions), but I was curious how was this perceived in terms of comfort and emotional stress.
Response 4: Thank you for pointing this out. We would like to clarify that, despite the use of blindfolding, all participants reported feeling comfortable, and none of them experienced stress or discomfort. Approximately every 5–7 minutes, the participants’ position was changed in order to examine the contralateral elbow. At these times, they were able to remove the blindfold and use their vision. Thus, the participants were not blindfolded continuously throughout the entire procedure. In addition, constant communication was maintained between the researchers and the participants for the purposes of the study, which further minimized any potential stress.
Reviewer 2 Report
Comments and Suggestions for Authors This study explored whether inertial measurement units can accurately and consistently measure elbow joint position sense during both active and passive tasks. Twenty healthy adults were tested with an IMU setup and a Biodex System 4 as the reference. For passive JPS the IMUs performed well, showing inter- and intra-rater reliability similar to the Biodex. Active JPS results were a bit less consistent. Bland–Altman analysis showed strong agreement between the two systems overall. The authors suggest that IMUs could be a valid, reliable, portable and low-cost option for elbow JPS testing, while noting that further testing in clinical and athletic settings is still needed. However, I have a few questions for the authors:- A key strength is the direct, synchronized comparison with the Biodex gold standard for the passive condition. This clear approach gives real weight to the claims of concurrent validity.
- The lower reliability seen in the active tests needs a closer look. The heavy Biodex lever may contribute, but it is unclear whether the drop is due to IMU limitations or the higher cognitive and motor demand of actively reproducing a joint angle. This seems more like a protocol issue than a sensor problem.
- The title and abstract refer to “elbow proprioception,” but the measurements were limited to static joint position sense. Because proprioception also includes kinesthesia and force sense, the claims throughout the paper should be tightened to reflect what was actually measured.
- All testing took place with participants fixed in a Biodex chair under controlled lab conditions. While this setup reduces confounding factors, it does not show how well the IMUs would perform in more natural or field settings, which is where portability matters most. The authors already highlight this as an important next step.
- Kendall’s coefficient of concordance is appropriate for non-normal data, but most biomechanics studies also report intraclass correlation coefficients. Adding ICC values alongside Kendall’s W would make it easier to compare with the wider literature.
- The paper mentions that a Kalman filter was used but gives no description of how it was implemented. A brief explanation would make the data processing clearer.
- Key details such as sampling rate, filtering methods, and how joint angles were calculated from IMU orientation data are missing. Without these, reproducing the results would be difficult.
- The IMU was placed on the forearm between the ulna and radius styloid processes. A short explanation of why this location was chosen and how it minimizes skin movement artifacts would strengthen the methods section.
- The authors emphasize low cost as an advantage but provide no numbers. A simple approximate cost comparison between the IMU system and the Biodex would make the argument for clinical adoption more convincing.
Author Response
Comment 1: A key strength is the direct, synchronized comparison with the Biodex gold standard for the passive condition. This clear approach gives real weight to the claims of concurrent validity.
Response 1: Thank you for your positive comment. We appreciate that.
Comment 2: The lower reliability seen in the active tests needs a closer look. The heavy Biodex lever may contribute, but it is unclear whether the drop is due to IMU limitations or the higher cognitive and motor demand of actively reproducing a joint angle. This seems more like a protocol issue than a sensor problem.
Response 2: Thank you for pointing this out. We agree with your comment. Accordingly, thanks to your suggestions we have provided in the discussion section these potential reasons for the substantially lower reliability observed in active JPS testing. The changes can be found on page 13, line 356-361.
Comment 3:The title and abstract refer to “elbow proprioception,” but the measurements were limited to static joint position sense. Because proprioception also includes kinesthesia and force sense, the claims throughout the paper should be tightened to reflect what was actually measured.
Response 3: Thank you for pointing this out. We agree that proprioception, in addition to joint position sense, also includes kinesthesia and force sense. We had previously clarified this at the beginning of the Discussion (page 11, lines 285-289). In order to not mislead readers what was actually measured, we made changes in the Materials and methods section. Term proprioception was changed to JPS (Joint position sense): in the beginning of paragraph 2.2, line 110 and in line 133. Moreover corrections were made in Figures 5 and 6 (previously 4 and 5) highlighting that we measured JPS. In this regard, as the whole Materials and methods section is clear, we consider that no further changes to the manuscript are necessary. While JPS does not encompass all aspects of proprioception, it is a fundamental and widely accepted component, reflecting proprioception.
Comment 4: All testing took place with participants fixed in a Biodex chair under controlled lab conditions. While this setup reduces confounding factors, it does not show how well the IMUs would perform in more natural or field settings, which is where portability matters most. The authors already highlight this as an important next step.
Response 4: Thank you for this comment. We appreciate that. Yes, controlled settings that allowed for a direct comparison between RSQ Motion sensors and the BIODEX system. Further research is needed to validate the performance of these inertial motion sensors in more diverse settings, particularly outside of the BIODEX chair. We already stated that in the Discussion section (page 13, lines 363-365 and 376-378).
Comment 5: Kendall’s coefficient of concordance is appropriate for non-normal data, but most biomechanics studies also report intraclass correlation coefficients. Adding ICC values alongside Kendall’s W would make it easier to compare with the wider literature.
Response 5: Thank you for pointing this out. We are aware that ICC is more commonly reported in biomechanics studies. However, ICC assumes normally distributed data, and our data samples exhibit a high degree of skewness, being highly non parametric. We implemented the Lilliefors test for normality, the Shapiro–Wilk test for normality, and the D’Agostino–Pearson test for normality. All these tests showed that our data do not follow normal distribution. Moreover, Q–Q (quantile–quantile) plots were inspected to visually confirm the high deviation from normality. Therefore, calculating ICC for our dataset would not be appropriate and could yield unreliable results. In the case of our study, comparing ICC values with those from other studies could be misleading and unreliable. For this reason, we believe that reporting ICC is not appropriate, and we opted to use Kendall’s coefficient of concordance, which is suitable for non-parametric data. We have additionally provided information about implemented tests used to check for distribution of our data: on page 7, lines 209-214.
Comment 6: The paper mentions that a Kalman filter was used but gives no description of how it was implemented. A brief explanation would make the data processing clearer.
Response 6: Thank you for pointing this out. The paper mentions a Kalman filter as an example of filters, not as the one used in our study. I addressed this inquiry to the sensor manufacturer. In response, I was informed that only the Madgwick complementary filter was used, which is specifically designed to estimate orientation from an accelerometer and gyroscope without requiring magnetic measurements. A Kalman filter–based approach typically requires magnetometer data to correct yaw drift and obtain an absolute directional reference. In our study, the magnetometer was intentionally disabled to avoid interference from the Biodex system. Changes can be found on page 2, lines 81-86.
Comment 7: Key details such as sampling rate, filtering methods, and how joint angles were calculated from IMU orientation data are missing. Without these, reproducing the results would be difficult.
Response 7: Thank you for pointing this out. We agree with your comment. Therefore, we addressed this question to the sensor manufacturer. Based on all the data that we obtained, we created a clear table summarizing the key RSQ Motion sensor specifications provided by the manufacturer. The changes can be found in Table 1, on page 4.
Comment 8: The IMU was placed on the forearm between the ulna and radius styloid processes. A short explanation of why this location was chosen and how it minimizes skin movement artifacts would strengthen the methods section.
Response 8: Thank you for pointing this out. We agree with your comment. Accordingly, we have addressed soft tissue artifacts in the context of the position used in our study in the Discussion section. The changes can be found on page 14, line 401-404.
Comment 9: The authors emphasize low cost as an advantage but provide no numbers. A simple approximate cost comparison between the IMU system and the Biodex would make the argument for clinical adoption more convincing.
Response 9: Thank you for pointing this out. We addressed this question to the sensor manufacturer. The response indicated that the cost of professional optometric systems and BIODEX ranges in the tens of thousands of euros and may, in some cases, approach one hundred thousand euros. In contrast, the cost of the basic sensor setup used in this study is approximately one thousand euros. This information has been included in the manuscript to make the argument for clinical adoption more convincing. The changes can be found on page 12, line 308-310.
Reviewer 3 Report
Comments and Suggestions for Authors
The manuscript is of some interest; it does not report major findings, but it is noteworthy because it aims to validate a simpler instrument. As the authors indicate, the objective of the paper is (lines 90–03):
“…was to assess the reliability of a protocol for evaluating elbow proprioception using inertial motion sensors and to validate the use of these sensors in comparison with a well-established, reference device – Biodex dynamometer.”
I suggest the authors report the following:
(lines 9–101) In the description of the sample they state the inclusion criteria, but in my view they should also report the sample characteristics (age, sex, etc.). This is essential, as it may bias the results.
Moreover, the sample is very small; please include an explicit statement of this limitation in the paper.
Line 197, Results. The reported correlations of 0.77 and 0.66, although significant, suggest substantial residual variability—in other words, a large proportion of variance remains unexplained. Therefore, this aspect should be examined in greater depth in the Discussion. The same applies to the correlations of 0.75 and 0.65 (pages 218–227); note that with the value 0.65 (line 222), nearly 60% of the variance remains unexplained. In sum, I suggest this be analyzed much more thoroughly in the Discussion.
If these revisions are made, in my opinion the paper can be published.
Author Response
Comments 1: (lines 9–101) In the description of the sample they state the inclusion criteria, but in my view they should also report the sample characteristics (age, sex, etc.). This is essential, as it may bias the results.
Response 1: Thank you for pointing this out. We agree with this comment. Therefore, we have added basic sample characteristics in materials and methods, under the ’’participants’’ paragraph. Changes can be seen on page 3, line 102-103.
Comments 2: Moreover, the sample is very small; please include an explicit statement of this limitation in the paper.
Response 2: Thank you for pointing this out. We agree with your comment. Accordingly, we have mentioned in the discussion section, under the limitations paragraph, that our study included a relatively small number of patients (20). We have also explained why this sample size was sufficient in the context of our objectives and methodology. The changes can be found on page 13, line 378-376.
Comments 3: Line 197, Results. The reported correlations of 0.77 and 0.66, although significant, suggest substantial residual variability—in other words, a large proportion of variance remains unexplained. Therefore, this aspect should be examined in greater depth in the Discussion. The same applies to the correlations of 0.75 and 0.65 (pages 218–227); note that with the value 0.65 (line 222), nearly 60% of the variance remains unexplained. In sum, I suggest this be analyzed much more thoroughly in the Discussion.
Response 3: Thank you for pointing this out. We agree with your comment. Accordingly, we have analyzed in the discussion section the potential reasons for the substantially lower reliability observed in active JPS testing. The changes can be found on page 13, line 356-361.
Reviewer 4 Report
Comments and Suggestions for Authors
The manuscript presents a comparison study between inertial measurement units (IMUs) and gold standard BIODEX for proprioception measurement. The experiment resuls show high correlation between those two sensor measurements in both active and passive trials with 20 able-bodied participants. The study is well-structured, easy to follow, and performs proper statistical analysis to evaluate reliability. The results provide useful insight for future research on proprioception assessment with portable sensors, particularly for the elbow joint. However, there are several concerns that should be clarified and addressed before publication.
Major Concerns:
- Section 2.2 lacks sufficient detail about the sensor specifications. Information such as sampling frequency, sensor composition, synchronization, filtering, and other technical parameters should be provided.
- The potential influence of sensor placement should be discussed, including human error during wearing, soft-tissue artifacts, and whether a calibration procedure was used before each experiment. If a standard calibration protocol exists, it should be described.
- The experimental design does not specify whether the order of reference angles was fixed or randomized. Order effects could bias the results, and this should be addressed or at least discussed.
- The rationale for selecting the specific target angle sets should be clarified. It is important to know whether they adequately represent the functional range of motion for proprioception.
- For passive trials, the speed of lever movement and the latency between button press and lever stop should be reported, as these factors may influence outcomes. Additionally, it should be stated whether participants completed practice rounds to familiarize themselves with the system before the actual data collection.
- For passive trials, sensing limb position while simultaneously preparing to press a button introduces potential cognitive load. The possible impact of this mental demand on proprioception performance should be discussed.
Minor Concerns:
- The practical implications of the study should be briefly but explicitly discussed in the abstract. Currently, the motivation appears to be framed mainly as a gap-filling exercise, rather than emphasizing why these findings matter for research or clinical practice.
- It is unclear what the authors mean by "Analyzed data was error of reproduction of joint position (ERJP)." The current wording is awkward and hard to follow.
- Figures, particularly Figures 4 and 5, should be improved. Currently, the text in the figure is too small with large areas of unused white space.
Author Response
Comment 1: Section 2.2 lacks sufficient detail about the sensor specifications. Information such as sampling frequency, sensor composition, synchronization, filtering, and other technical parameters should be provided.
Response 1:Thank you for pointing this out. We agree with your comment. Therefore, we addressed this question to the sensor manufacturer. Based on all the data that we obtained, we created a clear table summarizing the key RSQ Motion sensor specifications provided. The changes can be found in Table 1 on page 4.
Comment 2: The potential influence of sensor placement should be discussed, including human error during wearing, soft-tissue artifacts, and whether a calibration procedure was used before each experiment. If a standard calibration protocol exists, it should be described.
Response 2: Thank you for your comment. Concerning soft-tissue artifacts, the sensors were placed over prominent bony landmarks (the styloid processes of the ulna and radius). This ensured stable, precise positioning and minimized the thickness of underlying soft tissue, thereby reducing the likelihood of soft-tissue–related errors. Before each passive and active proprioception assessment, the device was recalibrated. The system is designed in such a way that initiating a new trial without performing calibration is not possible. The calibration was performed before each limb assessment. It involved placing the sensor on the ground and zeroing it in this position. We included this information in manuscript. The changes can be found on page 5, lines 142-144 and on page 14, lines 401-404.
Comment 3: The experimental design does not specify whether the order of reference angles was fixed or randomized. Order effects could bias the results, and this should be addressed or at least discussed.
Response 3: Thank you for pointing this out. The sequence of angle testing was fixed and determined by the order specified in the Biodex system protocol, which is considered the gold standard for JPS assessment. The reference angles were tested in the following order: 70°, 50°, and 110°. In our opinion, this order did not influence the study outcomes. However, we did not directly examine whether randomizing the sequence of angles would produce different results compared to the fixed order used in this study. The changes in manuscript regarding the order of angles can be found on page 6, lines 166-167, page 7, line 194 and on page 13, lines 365-369.
Comment 4: The rationale for selecting the specific target angle sets should be clarified. It is important to know whether they adequately represent the functional range of motion for proprioception.
Response 4: Thank you for your comment, The functional range of motion of the elbow joint is between 30° and 130° of elbow flexion, and the tested angles are within this range.
Comment 5: For passive trials, the speed of lever movement and the latency between button press and lever stop should be reported, as these factors may influence outcomes. Additionally, it should be stated whether participants completed practice rounds to familiarize themselves with the system before the actual data collection.
response 5: The angular velocity of the Biodex arm was 5 degrees per second. The manufacturer does not specify the latency between the button press and the lever stop. However, based on our observations, this response occurred instantaneously. The initial active and passive proprioception assessments included trial consisted of two single position records to allow the participant to become acquainted with the usage of the devices. Changes can be seen on 5. lines 148-150 and on page 7, line 187-187.
Comment 6: For passive trials, sensing limb position while simultaneously preparing to press a button introduces potential cognitive load. The possible impact of this mental demand on proprioception performance should be discussed.
Response 6: Thank you for pointing this out. Participants did not report any difficulties in simultaneously performing the proprioception tasks and pressing the button, and the researchers did not observe any such issues. Therefore, we assume that cognitive load did not significantly affect the study outcomes. It should be noted, however, that the participants were young, healthy individuals without conditions affecting cognitive function. Changes can be seen on page 13 lines 369-375.
Comment 7: The practical implications of the study should be briefly but explicitly discussed in the abstract. Currently, the motivation appears to be framed mainly as a gap-filling exercise, rather than emphasizing why these findings matter for research or clinical practice.
Response 7: Thank you for your insightful comment. We intentionally maintained a concise format which emphasizes methodological clarity and key outcomes. The practical, clinical relevance of our findings including their potential applications in clinical settings is discussed in detail in the Discussion on page 11-12, lines 304-341. We believe this placement allows for a more comprehensive and contextually appropriate explanation of the study’s implications without exceeding the abstract’s length or focus constraints.
Comment 8: It is unclear what the authors mean by "Analyzed data was error of reproduction of joint position (ERJP)." The current wording is awkward and hard to follow.
Response 8: Thank you for your comment. The analyzed data, referred to as the Error of Reproduction of Joint Position (ERJP), corresponds to the value obtained in each measurement. ERJP represents the accuracy with which participants were able to reproduce a specified joint position. It was quantified as the angular difference, in degrees, between the target angle and the angle reproduced by the participant. Subsequently, ERJP values were analyzed for intra-rater reliability, inter-rater. The additional explanation was added in the text. Change can be found on page 7 , lines 204-205.
Comment 9: Figures, particularly Figures 4 and 5, should be improved. Currently, the text in the figure is too small with large areas of unused white space.
Response 9 : Thank you for pointing this out. Figures 4 and 5 have been improved. The font size and the bars have been enlarged and emphasized to make the data presented in the figures easier for the reader to interpret. Figures 4 and 5 are now numbered as Figures 5 and 6, respectively, because an additional figure was added earlier in the text at the request of another reviewer.
Reviewer 5 Report
Comments and Suggestions for Authors
This paper investigates the validity and reliability of using inertial measurement units (IMUs) for assessing elbow joint proprioception, using the RSQ motion sensor system and the Biodex System 4. Twenty healthy participants underwent both active and passive joint position sense evaluations. The authors found that IMUs demonstrated good reliability and agreement with the Biodex system, especially in passive proprioception assessment, suggesting that IMUs are a practical and cost-effective alternative for clinical proprioception evaluation.
Major:
- The study oversimplifies elbow joint proprioception. IMUs can provide not only static responses but also dynamic ones. Parameters such as settling time and overshoot should be examined to enhance the paper’s contributions.
- As the authors stated that “the data did not follow a normal distribution,” they conducted statistical analyses. The definitions of these analyses should be clearly described so that readers can better understand and appreciate the conclusions.
- Please provide the data and the formula used for calculating inter-rater reliability, so that the (static/dynamic) results can be reproduced for further analysis.
- Figures 4 and 5 share the same captions despite representing different data. Please revise the captions to accurately reflect the content of each figure.
- In Figure 6, the y-axis label “SENSORS MINUS BIODEX” is not a scientifically appropriate expression. The error between the two sensor systems should be defined as an equation in the text. Additionally, the meaning of the two LogA lines should be clarified.
Minor:
- Section 3.2 appears to have a duplicated title: “Inter-Rater Reliability.” Please verify and correct if necessary.
Author Response
Comment 1: The study oversimplifies elbow joint proprioception. IMUs can provide not only static responses but also dynamic ones. Parameters such as settling time and overshoot should be examined to enhance the paper’s contributions.
Response 1: We appreciate the reviewer’s insightful comment regarding the potential of IMUs to provide dynamic parameters such as settling time and overshoot. However, the present study was designed specifically to evaluate static joint position sense (JPS), which represents one distinct aspect of proprioception. It is a classical and well-established method of assessing proprioception in the literature, particularly in the context of validating new devices. The experimental task and data processing were therefore optimized to assess error of reproduction of joint position (ERJP), rather than movement dynamics. We agree that including dynamic parameters could provide valuable additional insights into sensorimotor control, and we consider this an important direction for future research. In the Materials and Methods section, we emphasize that only JPS (Joint Position Sense) was measured. Moreover in the beginning of discussion we also state that JPS represents only one aspect of proprioception (page 11, lines 285-289) . Additionally, we have included Table 1 on page 4, which presents the all technical specifications of the sensors we could obtain from the sensor manufacturer.
Comment 2: As the authors stated that “the data did not follow a normal distribution,” they conducted statistical analyses. The definitions of these analyses should be clearly described so that readers can better understand and appreciate the conclusions.
Response 2: Thank you for your valuable comment. To assess the normality of the data distribution, we applied several statistical tests, including the Lilliefors test for normality, the Shapiro–Wilk test for normality, and the D’Agostino–Pearson test for normality. All of these tests indicated that the data did not follow a normal distribution. Additionally, skewness was evaluated, and Q–Q (quantile–quantile) plots were inspected to visually confirm the deviation from normality. These findings collectively justified the use of non-parametric statistical methods in our subsequent analyses. Changes can be seen on page 7, lines 209-214.
Comment 3: Please provide the data and the formula used for calculating inter-rater reliability, so that the (static/dynamic) results can be reproduced for further analysis.
Response 3: Thank you for your comment. In order to allow future reproduction of the results using the Kendall coefficient of concordance, we provided its formula, which is shown in Figure 4, on page 8. Unfortunately, we are unable to provide all the data for the concordance analysis, as they are subject to patient privacy regulations.
Comment 4: Figures 4 and 5 share the same captions despite representing different data. Please revise the captions to accurately reflect the content of each figure.
Response 4:Thank you for your comment. Figures 4 and 5 may appear similar in structure. However, they represent two distinct types of reliability: inter-rater reliability and intra-rater reliability, respectively. The bar labels and figure titles differ accordingly to reflect these separate analyses. To enhance clarity and visual quality, we have improved the resolution and presentation of both figures. . The font size and the bars have been enlarged. Due to adding Figure 4 according to your request, Figures 4 and 5 are now numbered as Figures 5 and 6, respectively.
Comment 5: In Figure 6, the y-axis label “SENSORS MINUS BIODEX” is not a scientifically appropriate expression. The error between the two sensor systems should be defined as an equation in the text. Additionally, the meaning of the two LogA lines should be clarified.
Response 5: Thank you for your comment. We have added an equation in the text to clearly define the error between the two measurement systems. Additionally, we have improved the quality and readability of Figure 6 and included an explanation of the LogA lines in the text. Changes can be seen on page 10, line 271-274. Figure 6 is now labeled as Figure 7, due to adding Figure 4 according to your request.
Round 2
Reviewer 4 Report
Comments and Suggestions for Authors
The authors addressed most of my comments. Just one more suggestion.
The figures in this paper are of very low quality. For example, Figure 4 must be improved, as low-resolution screenshots are not acceptable. In addition, equations should be typeset in the main text rather than included within figures.
Author Response
Comments 1:
The figures in this paper are of very low quality. For example, Figure 4 must be improved, as low-resolution screenshots are not acceptable. In addition, equations should be typeset in the main text rather than included within figures.
Response 1:
Dear Reviewer,
Thank you very much for your time and for the valuable comments regarding our manuscript. We have improved Figure 4, which is now presented in higher resolution and with clearer formatting. We have also enhanced the resolution of Table 1.
We believe that presenting Kendall’s coefficient of concordance in a figure rather than as inline equations in the text is more beneficial for readers, as it contains a long formula. This approach provides more space than embedding the equations within the text and allows us to better explain the components of the formula.
Reviewer 5 Report
Comments and Suggestions for Authors
The authors have addressed most of my previous comments; however, several major issues remain and should be resolved before the paper can be accepted.
- The equations in line 220 are poorly formatted. Please use standard scientific equation editors, such as Microsoft Equation Editor or MathType, to present them clearly.
- As shown in Figures 5 and 6, the standard deviations appear to be significantly larger than the mean differences. Please provide comments and explanations regarding this observation.
- The axes in Figure 7 should be labeled using scientific terminology. For example, “SENSORS – BIODEX” can be replaced with “error”, where the error is defined as:
error=θ_IMU−θ_Biodex
with θ_IMU and θ_Biodex representing the angles measured by the IMU and Biodex, respectively. Similarly, the x-axis label “MEAN (SENSOR + BIODEX)” should be revised to reflect a more scientific expression.”
Author Response
Comments 1: The equations in line 220 are poorly formatted. Please use standard scientific equation editors, such as Microsoft Equation Editor or MathType, to present them clearly.
Response 1: Thank you for pointing this out. We have improved resolution and formatting of equations in figure 4. The equations are now, in our opinion, sufficient to read and interpret, providing a clearer presentation of the mathematical relationships involved.
Comment 2: As shown in Figures 5 and 6, the standard deviations appear to be significantly larger than the mean differences. Please provide comments and explanations regarding this observation.
Response 2: Thank you for your comment. As we explain in the manuscript, our data did not follow a normal distribution (lines 209-214, page 7). Therefore, the standard deviation is noticeably higher, as our measurements deviated from the mean more than would be expected under a normal distribution.
Comment 3: The axes in Figure 7 should be labeled using scientific terminology. For example, “SENSORS – BIODEX” can be replaced with “error”, where the error is defined as:
error=θ_IMU−θ_Biodex
with θ_IMU and θ_Biodex representing the angles measured by the IMU and Biodex, respectively. Similarly, the x-axis label “MEAN (SENSOR + BIODEX)” should be revised to reflect a more scientific expression.”
Response 3: Thank you for pointing this out. Figure 7 has been revised according to the suggestion. In addition, more scientific equation expressions have been introduced in the text (lines 273-276, page 10).
Thank you very much for your time and for all the valuable comments regarding our manuscript.